



# Interpreting GEMS geostationary satellite observations of the diurnal variation of nitrogen dioxide (NO₂) over East Asia

Laura Hyesung Yang[1], Daniel J. Jacob[1,2], Ruijun Dang[1], Yujin J. Oak[1], Haipeng Lin[1], Jhoon Kim[3], Shixian Zhai[4], Nadia K. Colombi[2], Drew C. Pendergrass[1], Ellie Beaudry[1], Viral Shah[5,6], Xu Feng[1], Robert M. Yantosca[1], Heesung Chong[7], Junsung Park[7], Hanlim Lee[8], Won-Jin Lee[9], Soontae Kim[10], Eunhye Kim[10], Katherine R. Travis[11], James H. Crawford[11], Hong Liao[12]

[1] Harvard University, John A. Paulson School of Engineering and Applied Sciences, Cambridge, MA 02138, USA
[2] Harvard University, Department of Earth and Planetary Sciences, Cambridge, MA 01238, USA
[3] Yonsei University, Department of Atmospheric Sciences, Seoul 03722, South Korea
[4] Earth and Environmental Sciences Programme and Graduation Division of Earth and Atmospheric Sciences, Faculty of Science, The Chinese University of Hong Kong, Sha Tin, Hong Kong SAR, China
[5] Global Modeling and Assimilation Office, NASA Goddard Space Flight Center, Greenbelt, MD 20770, USA
[6] Science Systems and Applications, Inc., Lanham, MD 20706, USA
[7] Center for Astrophysics | Harvard & Smithsonian, Cambridge, Massachusetts 02138, USA
[8] Pukyong National University, Division of Earth Environmental System Science, Busan 48513, South Korea
[9] Environmental Satellite Center, National Institute of Environmental Research, Incheon 22689, South Korea
[10] Ajou University, Department of Environmental and Safety Engineering, Suwon 16499, South Korea
[11] NASA Langley Research Center, Hampton, VA 23666, USA
[12] Collaborative Innovation Center of Atmospheric Environment and Equipment Technology/Joint International Research Laboratory of Climate and Environment Change, School of Environmental Science and Engineering, Nanjing University of Information Science and Technology, Nanjing 210044, China

*Correspondence to*: Laura Hyesung Yang (laurayang@g.harvard.edu)

**Abstract.** Nitrogen oxide radicals (NO$_x$ ≡ NO + NO$_2$) emitted by fuel combustion are important precursors of ozone and particulate matter pollution, and NO$_2$ itself is harmful to public health. The Geostationary Environment Monitoring Spectrometer (GEMS), launched in space in 2020, now provides hourly daytime observations of NO$_2$ columns over East Asia. This diurnal variation offers unique information on the emission and chemistry of NO$_x$, but it needs to be carefully interpreted. Here we investigate the drivers of the diurnal variation of NO$_2$ observed by GEMS during winter and summer over Beijing and Seoul. We place the GEMS observations in the context of ground-based column observations (Pandora instruments) and GEOS-Chem chemical transport model simulations. We find good agreement between the diurnal variations of NO$_2$ columns in GEMS, Pandora, and GEOS-Chem, and we use GEOS-Chem to interpret these variations. NO$_x$ emissions are four times higher in the daytime than at night, driving an accumulation of NO$_2$ over the course of the day, offset by losses from chemistry and transport (horizontal flux divergence). For the urban core, where the Pandora instruments are located, we find that NO$_2$ in winter increases throughout the day due to high daytime emissions and increasing NO$_2$/NO$_x$ ratio from





entrainment of ozone, partly balanced by loss from transport and with negligible role of chemistry. In

summer, by contrast, chemical loss combined with transport drives a minimum in the $NO_2$ column at 13-14

local time. Segregation of the GEMS data by wind speed further demonstrates the effect of transport, with

$NO_2$ in winter accumulating throughout the day at low winds but flat at high winds. The effect of transport

can be minimized in summer by spatially averaging observations over the broader metropolitan scale, under

which conditions the diurnal variation of $NO_2$ reflects a dynamic balance between emission and chemical

loss.

## 1. Introduction

The Geostationary Environment Monitoring Spectrometer (GEMS) satellite instrument was launched

in February 2020 by the National Institute of Environmental Research (NIER) to observe air quality over

East Asia. GEMS is the first geostationary instrument directed at air quality and provides hourly column

measurements of several gases including nitrogen dioxide ($NO_2$) (J. Kim et al., 2020). $NO_2$ is part of the

nitrogen oxides ($NO_x \equiv NO + NO_2$) radical family, which is emitted by fuel combustion and whose

chemistry plays a critical role in driving ozone ($O_3$) and fine particulate matter ($PM_{2.5}$) formation. $NO_2$

itself is of concern as an air pollutant. Loss of $NO_x$ is by atmospheric oxidation by the hydroxyl radical

(OH) and ozone, resulting in a lifetime of a few hours in summer and about a day in winter (Shah et al.,

2020). The diurnal cycle of $NO_2$ measured from geostationary orbit offers unique information on the

emission, chemistry, and transport of $NO_x$. Here we interpret the GEMS observations with the GEOS-

Chem chemical transport model (CTM) to better understand the processes controlling this diurnal cycle.

Several studies have examined the diurnal variation of $NO_2$ in urban air using surface

concentrations from air quality networks. The data typically exhibit bimodal maxima in the morning around

7-9 local time (LT) and in the evening around 19-21 LT, including over Beijing and Seoul (Cheng et al.,

2018; H. Kim et al., 2020). This has been commonly attributed to high $NO_x$ emission during morning and

evening rush hours (Kendrick et al., 2015; Cheng et al., 2018), but urban $NO_x$ emission inventories show

little variation during daytime (Miao et al., 2020). Moutinho et al. (2020) found that the morning and




evening $NO_2$ maxima could be driven by shallow mixing depths, in contrast to the middle of the day where

surface heating drives deep mixing and defines the depth of the planetary boundary layer (PBL) in daily

contact with the surface. The PBL depth extends typically to 1-3 km altitude.

Ground-based measurements of $NO_2$ columns are available from the Pandora Sun-staring

spectrometer instrument network used for validating satellite observations (Herman et al., 2009; Kanaya et

al., 2014; Judd et al., 2020; Verhoelst et al., 2021). Column measurements integrate concentrations from

the surface to the top of the atmosphere and are therefore not directly sensitive to mixing depth. The

Pandora network consists mainly of urban sites, where the $NO_2$ column and its variability are mainly within

the PBL (Yang et al., 2023). The Pandora data from Seoul tend to show an increasing trend in the early

morning followed by flat concentrations over the rest of the daytime, with less diurnal variation than $NO_2$

concentrations in surface air (Crawford et al., 2021). Nearby sites can show different diurnal variations,

pointing to a major role of local transport in driving this variation (Chang et al., 2022; S. Kim et al., 2023).

Satellite observations of $NO_2$ from polar sun-synchronous low-earth orbit (LEO) have been made

since 1995 starting with the GOME instrument (Martin et al., 2002) but observe by design at a single time

of day. Several studies have combined observations from the SCIAMACHY or GOME-2 instruments

observing in the morning at 9-10 LT and the OMI instrument observing in the afternoon at 13-14 LT to get

some information on $NO_2$ diurnal variation. Boersma et al. (2008) found decreases from morning to

afternoon over urban regions that they attributed to photochemical loss, and increases from morning to

afternoon over tropical biomass burning regions that they attributed to a midday maximum in emissions.

Boersma et al. (2009) found that the urban morning-to-afternoon decrease was largest in summer and

absent in winter. Penn and Holloway (2020) found that $NO_2$ column ratios between morning and afternoon

were lower than surface $NO_2$ concentration ratios, as would be expected from deeper vertical mixing in the

afternoon. Ghude et al. (2020) found an important role for transport in driving morning-to-afternoon

variations in $NO_2$ columns over urban India.

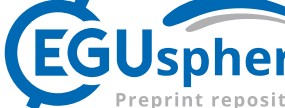

Here we analyze and compare the NO$_2$ diurnal cycles observed by GEMS over the Seoul and

Beijing metropolitan areas in winter and summer. We compare to the diurnal cycles observed by Pandora in

the urban cores and to simulations with the GEOS-Chem CTM. We use GEOS-Chem to separate and

quantify the roles of emission, chemistry, and transport in driving the NO$_2$ diurnal cycles observed from

GEMS over different spatial scales. This work provides a basis for more quantitative application of GEMS

observations as top-down information on NO$_x$ emissions, and more generally for interpreting the diurnal

cycle of NO$_2$ from geostationary orbit with application to the TEMPO instrument over North America

launched in April 2023 (Zoogman et al., 2017) and the Sentinel-4 instrument over Europe to be launched in

2024 (Gulde et al., 2017).

## 2. Observations and model

### 2.1 GEMS data

GEMS is an ultraviolet-visible instrument measuring back-scattered solar spectra at 300 – 500 nm

(J. Kim et al., 2020). It was launched in February 2020 in geostationary orbit at a longitude of 128.25°E. Its

native pixel resolution is $3.5 \times 8$ km$^2$ at Seoul, with variations in Level 2 product resolutions depending on

the pixel binning configurations. We use hourly total NO$_2$ slant column density from the GEMS L2 NO$_2$

version 2.0 product at $3.5 \times 8$ km$^2$ resolution for December-February (DJF) 2021/22 and June-August

(JJA) 2022 (NIER, 2023). The GEMS NO$_2$ algorithm uses differential optical absorption spectroscopy

(DOAS) to fit back-scattered solar spectra within the 432 – 450 nm range (J. Kim et al., 2020). This yields

the slant column density along the light path (L2 data). We use all GEMS L2 NO$_2$ version 2.0 data that pass

algorithm quality flag $\leq 112$, final algorithm flag $\leq 1$, solar zenith angle (SZA) < 70°, viewing zenith angle

(VZA) < 70°, and cloud fraction < 0.3 (Lee et al., 2020).

The vertical column density of NO$_2$ is obtained by dividing the slant column density by an air mass

factor (AMF) characterizing the photon path from the Sun down through the atmosphere and back up to the

instrument. The AMF depends on the viewing geometry and on the scattering properties of the atmosphere:

$$\text{AMF} = \text{AMF}_G \int_0^\infty w(z)S(z)dz \tag{1}$$



where AMF$_G$ is the geometric AMF defined by the solar zenith angle (SZA) and the satellite viewing angle

(VZA), $w(z)$ is the scattering weight that defines the instrument's sensitivity to NO$_2$ at altitude $z$, and $S(z)$

is a normalized vertical profile of NO$_2$ number density called the shape factor (Palmer et al., 2001).

Scattering weights are calculated with a radiative transfer model and typically increase with altitude

(Martin et al., 2002). The shape factor is usually estimated with a CTM.

Here we use scattering weights at 448 nm calculated for the application to a 435 – 461 nm spectral

window compiled as a look-up table dependent on SZA, VZA, relative azimuth angle (RAA), surface

albedo, cloud top pressure, and effective cloud fraction (R. Park and Kwon, 2020). We specify the shape

factor with local NO$_2$ concentrations from the GEOS-Chem simulation described in Section 2.3. In

simulations of observations from the KORUS-AQ aircraft campaign over Seoul, Yang et al. (2023) showed

that GEOS-Chem was successful in reproducing the observed diurnal variation in the vertical profile of

NO$_2$ as driven by PBL mixing, with important implications for the diurnal variation in the AMF. The

GEOS-Chem simulation extends vertically to the mesosphere, and we compute in this manner the AMF for

the whole atmosphere to obtain a total vertical column density. The contribution of the stratosphere to the

total NO$_2$ column in GEOS-Chem is only 9- 20% for the Seoul and Beijing metropolitan areas of interest

here. The PBL extending to 2 km altitude accounts for 95% of the tropospheric column over the Seoul

Metropolitan Area (Yang et al., 2023).

### 2.2 Pandora data

The Pandora instruments measure radiance at 280 – 525 nm (Herman et al., 2018) and fit total column

NO$_2$ (including the stratosphere). There are two Pandora sites in Seoul and one in Beijing (40.0°N,

116.4°E; O. Liu et al., 2023) for the 2021-2022 period. The two Pandora sites in Seoul are at Seoul

National University (Seoul – SNU; 37.5°N, 127.0°E) in the southern part of Seoul (M. Kim et al., 2021; S.

Park et al., 2018) and at Yonsei University (Seoul – YSU; 37.6°N, 126.9°E) in the northern part of Seoul (J.

Kim., 2017). We obtain the Pandora direct Sun data from the Pandonia global network (PGN, 2023) and

exclude data with a NO$_2$ flag of 12.



**2.3 GEOS-Chem model**

140  We use GEOS-Chem CTM version 13.3.4 ([https://doi.org/10.5281/zenodo.5764874](https://doi.org/10.5281/zenodo.5764874)) driven by

assimilated meteorological data from the Goddard Earth Observing System – Forward Processing (GEOS-

FP) with a horizontal resolution of $0.25° \times 0.3125°$ ($\approx 25\times25$ km$^2$) over East Asia ($24 – 52°N, 104 –$

$133°E$) and 3-hourly boundary conditions from a global GEOS-Chem simulation with $4° \times 5°$ resolution.

The model has 47 vertical levels including 14 vertical levels in the lower 2 km. Simulations were

145 conducted for DJF 2021/2022 and JJA 2022 with 6 months of initialization for each period.

  Aside from emissions (see below), the simulation is the same as previously described by Yang et al.

(2023) and features some modifications to the standard GEOS-Chem 13.3.4 to better reproduce KORUS-

AQ aircraft observations over Korea in May-June 2016. These include aerosol nitrate photolysis, volatile

chemical product (VCP) emissions and chemistry, and reduced HO$_2$ uptake by aerosol. Yang et al. (2023)

150 showed that the model could successfully simulate NO$_x$ chemistry during KORUS-AQ including the

diurnal variation of NO$_2$ vertical profiles affecting the diurnal variation of the AMF. This gives us

confidence in the application of GEOS-Chem normalized vertical profiles in Eq. (1) to compute AMFs for

application to the GEMS data.

  Simulations for 2022 require adjustment to NO$_x$ emissions beyond the most recent emission

155 inventories used in GEOS-Chem for China (MEIC for 2019; Zheng et al., 2021) and Korea (KORUSv5 for

2015; Woo et al., 2020). We apply for this purpose the surface NO$_2$ concentration trends for China from the

Ministry of Ecology and Environment (MEE) network (MEE, 2023) and for South Korea from the

AirKorea network (KEC, 2023). Mean 2022/2019 surface NO$_2$ concentration ratios in China are 0.91 in

DJF and 0.83 in JJA, and mean 2022/2015 values in Korea are 0.70 in DJF and 0.51 JJA, which are applied

160 to scale the anthropogenic NO$_x$ emissions.

  Several previous studies aside from Yang et al. (2023) have evaluated the GEOS-Chem simulation

of NO$_x$ over East Asia. R. Park et al. (2021) found that GEOS-Chem successfully reproduced the NO$_x$

vertical profiles observed during KORUS-AQ. Shah et al. (2020) found a good simulation of the



seasonality of OMI $NO_2$ over China and its long-term trend. M. Liu et al. (2018) found that $NO_2$ diurnal

variability at the MEE stations was well captured but the model was too low, as would be expected from

the urban nature of the sites.

### 2.4 Diurnal variation of $NO_x$ emissions

Figure 1 shows the diurnal cycle of $NO_x$ emissions used by GEOS-Chem in Beijing and Seoul.

MEIC for China provides monthly $NO_x$ emissions separately for the transportation, residential, industrial,

and power sectors while KORUSv5 separates mobile, area, and point sources. Neither inventory specifies

diurnal variations in emissions. In our work, we apply the diurnal pattern from X. Liu et al. (2019) for the

power sector and Miao et al. (2020) for other sources in the MEIC inventory. For KORUSv5 we apply the

diurnal pattern from X. Liu et al. (2019) for point sources, supported by results from Bae et al. (2021), and

the industrial daily pattern from Miao et al. (2020) for area sources. We estimate the diurnal variation of

mobile sources in KORUSv5 using hourly Seoul Transport Operation and Information Services (TOPIS,

2023) data on weekday total traffic and construction equipment activity.

Figure 1 shows that emissions are dominated by industrial and transport sources in Beijing, and by

mobile (transport) sources in Seoul. Both sectors show a broad maximum between 7 and 18 LT that defines

the overall diurnal cycle of emissions and is similar in winter and summer. There are no significant rush

hour peaks in transport emissions, suggesting that the surface $NO_2$ maxima observed in early morning and

evening are driven more by shallow mixing depths (Moutinho et al., 2020). Total $NO_x$ emission in Beijing

in winter is 30% greater than in summer, driven by the industrial source and possibly due to workplace

heating. There is less seasonal variation in Seoul where mobile sources are the largest emitters.

### 3.    Intercomparison of total $NO_2$ columns

Figure 2 shows the total $NO_2$ columns over eastern China and South Korea observed by GEMS and

simulated by GEOS-Chem during DJF 2021/22 and JJA 2022. The GEMS data are mapped on the 0.25° ×

0.3125° GEOS-Chem grid. The yellow box delineates the Seoul Metropolitan Area (SMA). The zoomed-in

black boxes are Beijing and Seoul and the white boxes are the city centers where Pandora stations are



located. The NO$_2$ plume location in winter is shifted south due to the prevailing winds and the long NO$_x$

lifetime (Seo et al, 2021). We see from Figure 2 that GEMS and GEOS-Chem have consistent spatial

distributions and backgrounds, but GEOS-Chem over polluted regions is generally higher than GEMS

except for Seoul in winter.

Figure 3 further intercompares GEOS-Chem and GEMS using the Pandora stations in Beijing and

Seoul as evaluation metric. GEMS and GEOS-Chem reproduce the diurnal and day-to-day variability

observed by Pandora in DJF ($R^2 = 0.87$-$0.90$) and JJA ($R^2 = 0.77$-$0.79$). NO$_2$ column magnitudes also agree

well with Pandora in winter, with linear regression slopes of 0.94 for GEMS and 0.90 for GEOS-Chem.

Summer shows larger biases reflecting differences between the SNU and YSU Pandora sites that cannot be

resolved at the $0.25° \times 0.3125°$ resolution of GEOS-Chem (there are few observations at the Beijing site in

JJA). YSU is more polluted than SNU, which is in a mountainous area more remote from emissions.

Overall, comparison to Pandora supports the diurnal and day-to-day variability seen in the GEMS and

GEOS-Chem data. The rest of our analysis focuses on the diurnal variability.

### 4.  Diurnal variation of NO$_2$ columns on the urban scale

We start with an urban core analysis focusing on the white boxes shown in Figure 2 for Beijing and

Seoul, representing single $0.25° \times 0.3125°$ GEOS-Chem grid cells where the Pandora stations are located.

Scatterplot comparisons between GEOS-Chem, GEMS, and Pandora for these grid cells were shown in

Figure 3. Figure 4 shows the diurnal variation of the total NO$_2$ column observed from GEMS and Pandora

and simulated by GEOS-Chem in Beijing. GEOS-Chem results are shown as averages for all days and for

the subset of days when GEMS observations are available (generally limited by cloud cover). Wintertime

NO$_2$ in all three datasets is flat from 10 to 11 LT and then increases from 11 to 14 LT. Summertime NO$_2$

decreases from 8 LT to a minimum at 13-14 LT and then increases to 16 LT, consistent between GEOS-

Chem and GEMS. Pandora observations in the summertime are too limited to show.

We used the GEOS-Chem budget tendency diagnostic to understand the drivers of the diurnal

variation in NO$_2$ columns. This diagnostic tracks the mean mass-weighted changes of column



concentrations after each model operation for any selected horizontal domain, vertical column, and time

period. We focus on the PBL column conservatively defined as extending to 3 km altitude after verifying

that altitudes higher than 3 km make negligible contributions to diurnal changes in the total model column.

Within the PBL column we consider the budget of $NO_2$ as that of $NO_x$ ($\equiv NO + NO_2 + NO_3 + 2N_2O_5 +$

$HONO + HNO_4 + ClNO_2$) multiplied by the local $NO_2/NO_x$ PBL column concentration ratio. This

eliminates from the budget the fast interconversion reactions within the $NO_x$ family and provides a more

useful budget perspective. The $NO_x$ family is mainly contributed by NO and $NO_2$, and the main

interconversion reactions defining the $NO_2/NO_x$ ratio are

$$NO + O_3 \rightarrow NO_2 + O_2 \qquad\qquad (R1)$$
$$NO + HO_2 \rightarrow NO_2 + OH \qquad\qquad (R2)$$
$$NO_2 + h\nu \xrightarrow{+O_2} NO + O_3 \qquad\qquad (R3)$$

The differential net budget tendency for the PBL $NO_2$ column ($\Omega_{NO2}$) can then be related to that of $NO_x$

($\Omega_{NOx}$) as

$$\left[\frac{\partial \Omega_{NO_2}}{\partial t}\right]_{net} = \alpha(t)\left[\frac{\partial \Omega_{NO_x}}{\partial t}\right]_{net} \quad (2)$$

with

$$\left[\frac{\partial \Omega_{NO_x}}{\partial t}\right]_{net} = \left[\frac{\partial \Omega_{NO_x}}{\partial t}\right]_{emission} + \left[\frac{\partial \Omega_{NO_x}}{\partial t}\right]_{chemistry} + \left[\frac{\partial \Omega_{NO_x}}{\partial t}\right]_{transport} \quad (3)$$

and where $\alpha(t) = \Omega_{NO_2}/\Omega_{NO_x}$ is the $NO_2/NO_x$ PBL column ratio. The terms on the right-hand side of Eq. (3)

are updated by GEOS-Chem over its operator splitting time steps and are archived in the budget diagnostic

as spatial and temporal averages. $NO_x$ dry deposition is included in the emission operator, but its

contribution is very small (Shah et al., 2020).

The second row of Figure 4 shows the different $NO_x$ budget terms from Eq. (3) over hourly time

steps, with the net tendency as the left-hand-side term. The third row shows the $NO_2/NO_x$ PBL column

molar ratios in GEOS-Chem. The GEOS-Chem diurnal variation in the $NO_2$ column in the first row reflects

the net $NO_x$ tendency combined with the $NO_2/NO_x$ ratio. In this manner, we see that the increase in the $NO_2$

column over the course of the day in winter reflects the dominant effect of daytime emissions, four times



higher than at night and leading to $NO_x$ accumulation. Chemical loss is slow in winter and transport (flux

divergence) is the main loss term. The flat trend of the $NO_2$ column from 10 to 11 LT corresponds to the

diurnal minimum of the $NO_2/NO_x$ ratio. This ratio increases over the rest of the day as the mixed layer

deepens and the freshly emitted NO is exposed to higher ozone concentrations. The increase in the ratio

contributes to the increase in the $NO_2$ column. The $NO_2$ column in GEOS-Chem thus peaks at 18 LT.

During the night, the $NO_x$ emission decreases and the loss from transport leads to decrease in the total $NO_2$

column. The $NO_2/NO_x$ ratio at night is only 0.55 mol mol$^{-1}$, despite no $NO_2$ photolysis, because of

sustained NO emission and the slow rate of the NO + $O_3$ reaction (low ozone and low temperatures).

        The opposite diurnal variation of $NO_2$ in summer reflects weaker daytime emission of $NO_x$ and

stronger chemical loss as shown by the GEOS-Chem budget analysis. Even though the emission term

remains larger than the chemical loss term, there is also a negative transport term because upwind

emissions are much lower. Without the transport loss term, the $NO_2$ column in summer would still increase

over the course of the day. The chemical loss of $NO_x$ peaks at 11-12 LT and then weakens, reflecting the

noon maximum of OH concentrations combined with the decreasing $NO_2$ concentration, and explaining the

slow recovery of the $NO_2$ column in the afternoon. The $NO_2/NO_x$ ratio is higher in summer than in winter

and shows little variation during the daytime, reflecting the higher concentrations of $O_3$ and $HO_2$ radicals

offsetting the effect of $NO_2$ photolysis. The daytime $NO_2/NO_x$ ratio averages 0.75 mol mol$^{-1}$ in summer, as

compared to 0.5 mol mol$^{-1}$ in winter, contributing to the seasonality of $NO_2$ seen from space.

        Figure 5 shows the same as Figure 4 but for Seoul. The two Pandora stations show differences in

$NO_2$ columns, particularly in summer, as previously shown in Figure 2. They also show some differences in

diurnal variation, particularly in winter, which we similarly attribute to local effects such as different

emissions, wind speeds, and geography that cannot be resolved at 25-km resolution. The diurnal variations

of GEMS and GEOS-Chem agree to within the ranges defined by the Pandora data. $NO_2$ columns in winter

increase from 10 to 12 LT as in Beijing but then flatten in the afternoon, which we attribute in GEOS-

Chem to stronger winds. $NO_2$ columns in summer show an increase from 8 to 10 LT, unlike in Beijing,





because of larger emissions initially overwhelming the chemical loss term. There follows a decrease until

13-14 LT and a recovery in the later afternoon, similar to Beijing and driven by the same factors.

**5.   Separating the influences of emission, chemistry, and transport**

We showed in Section 4 that the diurnal variation of the $NO_2$ column observed by GEMS on the

urban scale reflects a balance between emission and transport in winter, and the added influence from

chemical loss in summer. This suggests that emission and chemical loss can be independently inferred from

the GEMS observations in winter and summer if the role of transport can be quantified. The transport term

can be represented with a CTM in an inversion framework (Cooper et al., 2017), but simple quantification

of the transport term on the urban scale can also be done from knowledge of the wind speed with a mass

balance approach (Jacob et al., 2016).

Figure 6 illustrates the sensitivity of the $NO_2$ diurnal variation to wind speed in the wintertime

GEMS observations over Seoul when chemical loss is a negligible term. Here the data are segregated by

hourly wind speed (6 m s$^{-1}$) at 850 hPa from NASA GEOS-FP meteorology. The diurnal variations are very

different at high and low wind speed, and consistent between GEMS and GEOS-Chem. At high wind

speed, the $NO_2$ column shows little diurnal variability because emission is balanced by transport. At low

wind speed, $NO_2$ accumulates throughout the day because the transport term is weaker and does not keep

up with emissions. But the transport term is not negligible even at low wind speed; for a 3 m s$^{-1}$ wind, the

ventilation time scale for the 25-km urban domain is only 2 hours. This also explains why the transport

term remains important in summer (Figures 4 and 5), as the timescale for $NO_x$ chemical loss is about 6

hours (Shah et al., 2020).

One can reduce the effect of transport by spatial averaging over a large domain, thereby increasing

the ventilation timescale. Figure 7 shows the average diurnal pattern of the $NO_2$ column observed by

GEMS and simulated by GEOS-Chem on the ≈150-km scale of the SMA (Figure 2) in winter and summer.

Again, the diurnal variations observed by GEMS and simulated by GEOS-Chem are consistent. $NO_2$

columns in winter increase over the course of the day in a more regular manner than on the 25-km urban



scale (Figure 5) because the transport loss term is steadier and responds mainly to the change in the $NO_2$

column. The chemical loss term is not negligible, unlike on the urban scale, because its timescale of 24

hours is comparable to that of transport.

We see from Figure 7 that the transport term can be successfully marginalized on the scale of the

SMA in summer because the chemical loss term is faster. The resulting SMA diurnal pattern of the total

$NO_2$ column is consistent with that of Seoul (Fig. 5b) but with a flatter shape and the early-afternoon

minimum now driven mainly by chemistry. The amplitude is greatly dampened because emissions are five

times weaker when averaged over the SMA regional domain, and because the chemical loss integrates over

the residence time within the domain.

### 6. Conclusions

We used the GEOS-Chem model to interpret the diurnal variation of $NO_2$ columns observed from

the GEMS geostationary instrument and Pandora ground-based spectrometers over East Asia in December-

January-February (DJF) 2021/22 and June-July-August (JJA) 2022. This was motivated by the need to

understand the unique information offered by hourly geostationary satellite observations on the budget of

$NO_x$ through the contributions of emissions, chemistry, and transport to the diurnal cycle of $NO_2$.

The GEOS-Chem model used in this work had previously shown successful simulation of the $NO_2$

vertical profile and its diurnal variation over Seoul in the KORUS-AQ aircraft campaign, resolving the

diurnal dependence of the air mass factor (AMF) that contributes to the diurnal variation of $NO_2$ observed

from space. Here we used the diagnostic budget capability in GEOS-Chem to isolate the contributions of

$NO_x$ emissions, chemistry, transport, and the $NO_2/NO_x$ column ratio to the diurnal cycle of $NO_2$ columns.

We also updated $NO_x$ emissions to 2022 including their diurnal variations. $NO_x$ emissions for Beijing and

Seoul are a factor of four higher in the daytime than at night, reflecting mobile and industrial sources, and

show little variation during the daytime hours. We focused on simulation of the total atmospheric $NO_2$

column rather than the tropospheric column, taking advantage of the stratospheric capability in GEOS-



Chem, and to avoid errors in the definition of the tropopause. Diurnal variation of the $NO_2$ atmospheric

column is mainly determined by the planetary boundary layer (PBL) up to 3 km altitude.

We investigated the diurnal variation of the $NO_2$ column at the 25-km urban scale over Beijing and

Seoul. GEMS, Pandora, and GEOS-Chem show similar variability and diurnal variations. $NO_2$ columns in

winter increase over the course of the daytime hours, reflecting accumulation from high daytime emissions

offset by loss from horizontal transport (flux divergence), and further enhanced by increase in the $NO_2/NO_x$

over the course of the day as ozone is entrained in the growing mixed layer. Chemical loss of $NO_x$ in winter

is too slow to play a significant role in the observed diurnal variation. In summer, by contrast, $NO_2$ columns

decrease from 10 to 14 local time (LT) because of $NO_x$ photochemical oxidation compounding the loss

from transport. The $NO_2$ column would still increase throughout the day in summer as in winter were it not

for the loss from transport.

Our results indicate that the diurnal variation of $NO_2$ column observed from geostationary orbit can

be used to quantify urban $NO_x$ emissions in winter, and chemical loss in summer, but the transport term

must be accounted for. This can be done by simple mass balance using knowledge of the local wind speed

or by an inversion with a full CTM.

We further examined the importance of transport for interpreting the diurnal variation in the GEMS

urban $NO_2$ data by segregating the Seoul data by wind speed. In winter, the low-wind GEMS data ($< 6$ m s$^{-1}$) show steady rise of $NO_2$ over the course of the day while the high-wind data ($\geq 6$ m s$^{-1}$) show flat diurnal

variation, consistent with the GEOS-Chem the model. Transport plays an important role in the $NO_x$ budget

in both cases but cannot keep up with the high daytime emissions in the low-wind case.

We examined whether the role of transport in the diurnal variation of the urban $NO_2$ column could

be reduced by spatial averaging of the data over the 150-km regional scale of the Seoul Metropolitan Area

(SMA). The SMA data in winter show a steady increase over the daytime hours due to emissions, but the

transport term remains the major sink of $NO_x$. The SMA data in summer show negligible loss from



transport in daytime because chemical loss term is much faster, but the diurnal amplitude is weak because of diluted emissions and long residence times for the air over the regional domain.

Our conclusions regarding the interpretation of the diurnal variation of $NO_2$ columns observed by GEMS can be extended to other instruments of the geostationary air quality constellation, such as TEMPO over North America, launched in April 2023 and Sentinel-4 over Europe, scheduled for launch in 2024. This work further lays the groundwork for use of GEOS-Chem in inversions of the geostationary satellite data to infer $NO_x$ emissions.

**Code Availability**

The model code used in this work is available at https://doi.org/10.5281/zenodo.5764874 (The International GEOS-Chem User Community, 2021).

**Data availability**

The GEMS L2 $NO_2$ v2.0 slant column data can be obtained with a request to the NIER GEMS team (NIER,
2023). The total $NO_2$ columns from Pandora are available from the Pandonia Global Network website (http://pandonia-global-network.org; PGN, 2023). The surface $NO_2$ data over China are available from https://quotsoft.net/air/ (MEE, 2023). The surface $NO_2$ data over South Korea are from AirKorea website (https://www.airkorea.or.kr; KEC, 2023). The Seoul hourly traffic count data is available from the Seoul Transport Operation and Information Services website (https://topis.seoul.go.kr/; TOPIS, 2023).

**Acknowledgements**

This material is based upon work supported by the National Science Foundation Graduate Research Fellowship under grant no. DGE 2140743.

**Author contribution**

The original draft preparation was done by LHY, with review and editing by DJJ, KRT, JHC, HL, and JK.
DJJ contributed to project conceptualization. Modeling was done by LHY with additional support from HPL, NKC, SZ, VS, EB, XF, RMY, and DCP. The formal analysis was conducted by LHY with additional support from DJJ, RD, YJO, HC, JP, HLL, WJL, SK, EK, KRT, and JHC.



**Competing Interests**

The contact author has declared that none of the authors has any competing interests.

**Financial support**

This research was supported by the Samsung Advanced Institute of Technology and the Nanjing University of Information Science and Technology.

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

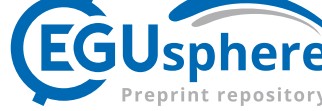

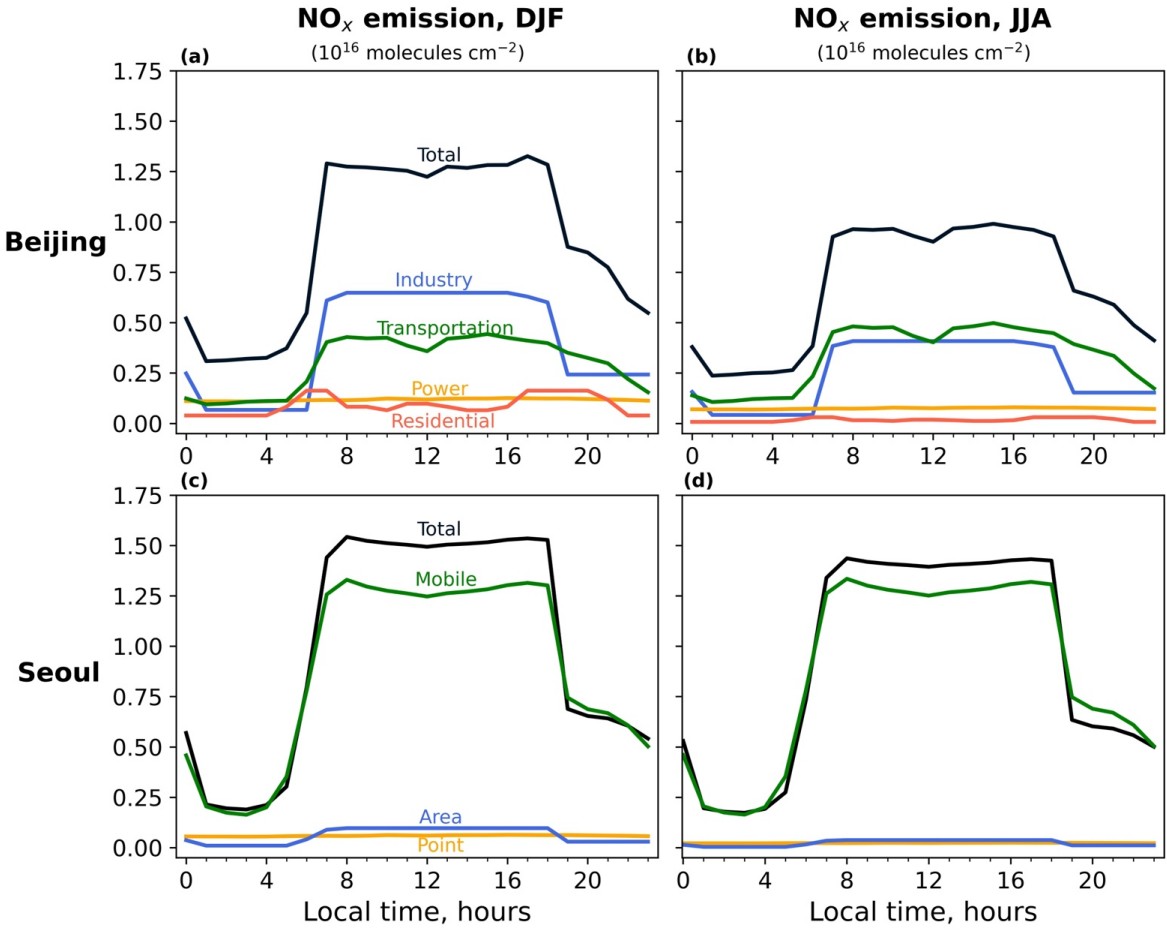

**Figure 1.** Diurnal variation of NO$_x$ emissions in Beijing and Seoul for DJF 2021/2022 and JJA 2022. Local time is Chinese Standard Time (CST) for Beijing and Korean Standard Time (KST) for Seoul. Solar noon is at 12:08 – 12:27 CST in Beijing and 12:21 – 12:45 KST in Seoul. Values are for the white boxes in Figure 2. Different colors represent different sectors, and the black line shows the total emission.



**Figure 2.** Total NO$_2$ columns over East Asia retrieved by GEMS and simulated by GEOS-Chem. The data are 3-month averages for December-July-February (DJF) 2021/22 and June-July-August (JJA) 2022 on the 0.25° × 0.3125° GEOS-Chem nested grid. The yellow rectangle delineates the Seoul Metropolitan Area (SMA; 36.6-38.1°N, 126.4-128.3°E). The zoomed-in plots show Beijing and Seoul, and the white boxes are the 0.25° × 0.3125° urban cores where the Pandora stations are located (black circles). Scales are different for DJF and JJA.





**Figure 3.** Intercomparison of GEOS-Chem, GEMS, and Pandora $NO_2$ columns for the Pandora sites in
Beijing and Seoul. The Figure shows scatterplots of daytime hourly data for DJF 2021/2022 and JJA 2022.
GEMS is mapped on the 0.25° × 0.3125° GEOS-Chem grid. Coefficients of determination ($R^2$) and
reduced-major axis linear regressions are shown. The 1:1 line is dashed. The Beijing Pandora site has
limited observations in JJA.

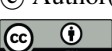

**Figure 4.** Diurnal variation of total $NO_2$ column and driving processes in Beijing. The first row shows the average $NO_2$ columns observed by GEMS and Pandora, and simulated by GEOS-Chem, in DJF 2021/22 and JJA 2022 for the $0.25° \times 0.3125°$ GEOS-Chem grid cell in the urban core where the Pandora station is located (white box in Figure 2). GEMS observations are available for the hours indicated by symbols. GEOS-Chem results for the full diurnal cycle are shown as averages for all days and for the subset of days when GEMS data are available (generally limited by cloud cover). Pandora data are not shown for JJA due to a limited number of observations (Figure 3). The second row shows the hourly tendencies in the GEOS-Chem $NO_x$ budget (averaged for all days) for the planetary boundary layer (PBL) conservatively defined as extending up to 3 km altitude. The tendencies describe the contributions from individual processes to the $NO_x$ budget as given by Eq. (3), with $NO_x$ defined as $NO_x \equiv NO + NO_2 + NO_3 + 2N_2O_5 + HONO + HNO_4 + ClNO_2$. The third row shows the PBL $NO_2/NO_x$ column molar ratio in GEOS-Chem.




**Figure 5.** Same as Figure 4 but for Seoul.

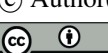



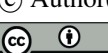

**Figure 6.** Same as Figure 4 but for DJF 2021/22 in Seoul with data segregated by wind speed. Segregation threshold is 6 m s$^{-1}$ for the 850 hPa hourly wind speed in the NASA GEOS-FP meteorological data used as input to GEOS-Chem.



**Figure 7.** Same as for Figure 4 but for the Seoul Metropolitan Area (SMA; 36.6-38.1°N, 126.4-128.3°E) corresponding to the yellow box in Figure 2.

605