# Peer review of "Interpreting GEMS geostationary satellite observations of the diurnal variation of nitrogen dioxide (NO2) over East Asia"

_EGUsphere, 2023_

## Community Comment (CC1)

Thanks for your interesting work. I have a few questions and would greatly appreciate your assistance:

1. In the Introduction section's second and third paragraphs, I'm slightly confused by the mixed use of the terms "mixing depths" and "PBL depths". Are they referring to the same thing?

2. In lines 100-105, you mention that the native pixel resolution is 3.5x8 km² in Seoul. Could you please clarify what the native pixel resolution is outside of Seoul?

3. In Equation 1, is there a specific formula for $AMF_G$ that depends on SZA and VZA?

4. In lines 119 to 121, would scattering weights also be dependent on altitude $z$?

5. In Section 2.2, could you possibly provide a bit more detail about the Beijing sites, similar to the descriptions you provided for the Seoul sites?

6. In lines 185 to 195, the location of the $NO_2$ plume is not clearly labelled on the figure. Could you please clarify this? Furthermore, while Figure 2 is quite informative, I believe it could be enhanced by adding a third column that shows the bias, calculated as the difference between the model and the observations. Would that be possible?

7. In lines 195 to 200, I noticed that measurements from the two Pandora sites were not aggregated for the GEOS-Chem grid, unlike what was done for the GEMS data. Could you please explain why this is the case? Alternatively, may you also compare the two Pandora sites to the non-aggregated GEMS data?

8. I'm trying to understand the methodology used to derive the net change of $NO_2$ from the net change of $NO_x$. Considering two time points $t1$ and $t2$, with corresponding column concentrations for NO2 (denoted as $NO_2^{t1}$ and $NO_2^{t2}$), $NO_x$ (denoted as $NO_x^{t1}$ and $NO_x^{t2}$), and their ratios (denoted as $\alpha^{t1}$ and $\alpha^{t2}$), we can calculate the rate of change of $NO_2$ and $NO_x$ as
$\frac{\partial NO_2}{\partial t}\Big|_{t=t2} = \frac{NO_2^{t2}-NO_2^{t1}}{t2-t1}$ and $\frac{\partial NO_x}{\partial t}\Big|_{t=t2} = \frac{NO_x^{t2}-NO_x^{t1}}{t2-t1}$, respectively. As far as I am concerned, we can only transform the former into something like
$\frac{\partial NO_2}{\partial t}\Big|_{t=t2} = \frac{\alpha^{t2}\times NO_x^{t2}-\alpha^{t1}\times NO_x^{t1}}{t2-t1}$, but this doesn't seem equal to
$\alpha^{t2} \times \frac{NO_x^{t2}-NO_x^{t1}}{t2-t1} = \alpha^{t2} \times \frac{\partial NO_x}{\partial t}$ unless we assume $\alpha^{t1} = \alpha^{t2}$. Is this assumption being made? Alternatively, can the GEOS-Chem model directly provide $\frac{\partial NO_2}{\partial t}$ and its individual components? I apologize if I've misunderstood any aspects of your methodology.

9. As you can derive the net change of $NO_2$ from the net change of $NO_x$, would it be more straightforward in Section 4 to directly analyse the individual

components of the net change of $NO_2$ rather than combining the net change of $NO_x$ with $NO_2/NO_x$ ratios to facilitate the analyses? Are you doing the latter way because the $NO_2/NO_x$ ratios may have implications for something like $O_3$ entrainment. Nonetheless, I am assuming that both GEMS and GEOS-Chem can provide $O_3$.

10. In lines 249 and 250, why was the presence of the negative transport term linked to the upwind emissions being much lower?

11. In lines 250 to 255, it appears that you're discussing the variations of $NO_2$ alongside the variations of $NO_x$. Do you have evidence for the maximum concentration of OH at noon, or is this a generally accepted knowledge that's prescribed in the GEOS-Chem model?

12. In line 258, it appears that the discrepancy between the two Pandora sites is more clearly illustrated in Figure 3 rather than in Figure 2.

13. In lines 260 and 261, could you clarify what range is defined by the Pandora data? Additionally, could you explain how the diurnal variations observed by GEMS and simulated by GEOS-Chem agree within this defined range?

14. In lines 270 and 271, are you suggesting that if the transport term can be quantified by simple methods, satellite observations could directly indicate the role of emission and chemistry without the need for the GEOS-Chem model? However, the follow-on analysis in Figure 6 still have the transport term under different conditions. Similarly, in lines 293 and 294, are you suggesting that on a regional scale, the tranport term can be marginalized (minimized), leading us to interpret that satellite observations primarily reflect the contributions from emissions and chemistry? How would this statement apply to other regions and periods?

15. In lines 281 and 282, what is the source of the numerical relationship between wind speed and the ventilation time scale?

16. In Section 5, I'm curious whether the results for the SMA metropolitan area were obtained simply by averaging the results from the grids that belong to the SMA metropolitan area?

17. For Figures 4 — 7, the points on the lines in the first row correspond to exact hours (e.g., 8:00, 9:00, etc.), while the points on the lines in the second and third rows correspond to the half-hour marks (8:30, 9:30, etc.). Is this an intended behavior?

---

## Author Comment (AC1)

Our response can be found in blue font.

**Response to Comments by Community Commentator #1 (Fei Yao)**

Thanks for your interesting work. I have a few questions and would greatly appreciate your assistance:
We thank the community commentator #1 for reviewing our paper and providing constructive feedback. Our response to the questions is as follows:

1. In the Introduction section's second and third paragraphs, I'm slightly confused by the mixed use of the terms "mixing depths" and "PBL depths". Are they referring to the same thing?
We removed the confusion in the text. The line that refers to the PBL depth is revised as follows: "… in contrast to the middle of the day and afternoon hours when surface heating maximizes the mixing depth. This diurnal maximum in mixing depth defines the planetary boundary layer (PBL) in daily contact with the surface.

2. In lines 100-105, you mention that the native pixel resolution is 3.5x8 km² in Seoul. Could you please clarify what the native pixel resolution is outside of Seoul?
We revised the sentence to be "We use hourly total $NO_2$ slant column density from the GEMS L2 $NO_2$ version 2.0 product at native $3.5 \times 8$ km$^2$ resolution for December-February (DJF) 2021/22 and June-August (JJA) 2022 (NIER, 2023)."

3. In Equation 1, is there a specific formula for $AMF_G$ that depends on SZA and VZA?
The description for equation 1 is revised as "…$AMF_G$ is the geometric AMF defined by the solar zenith angle (SZA) and the satellite viewing angle (VZA) as $AMF_G = \sec(SZA) + \sec(VZA)$, …"

4. In lines 119 to 121, would scattering weights also be dependent on altitude z?
We have the following line: "Scattering weights are calculated with a radiative transfer model and increase with altitude (Martin et al., 2002; Yang et al., 2023)."

5. In Section 2.2, could you possibly provide a bit more detail about the Beijing sites, similar to the descriptions you provided for the Seoul sites?
We added the following line in Section 2.2: "The Beijing site is located on the north side of Beijing and a more detailed description is in O. Liu et al. (2024)."

6. In lines 185 to 195, the location of the NO2 plume is not clearly labelled on the figure. Could you please clarify this? Furthermore, while Figure 2 is quite informative, I believe it could be enhanced by adding a third column that shows the bias, calculated as the difference between the model and the observations. Would that be possible?
We clarify in the text as follows: "The maximum $NO_2$ concentrations are in the city centers in summer, but are shifted to the south in winter due to the prevailing winds and the long $NO_x$ lifetime (Seo et al, 2021)." We appreciate the suggestion but it would overcomplicate Figure 2. We believe the readers can find the difference in magnitude for GEMS and GEOS-Chem in Figure 3.

7. In lines 195 to 200, I noticed that measurements from the two Pandora sites were not aggregated for the GEOS-Chem grid, unlike what was done for the GEMS data. Could you please explain why this is the case? Alternatively, may you also compare the two Pandora sites to the non-aggregated GEMS data?
We added the following line to the main text: "Previous GEMS evaluation with Pandora at the native pixel resolution of GEMS was presented by S. Kim et al. (2023). Here we conduct the evaluation on the coarser $0.25° \times 0.3125°$ GEOS-Chem grid as most relevant for our work."

8. I'm trying to understand the methodology used to derive the net change of NO2 from the net change of NOx. Considering two time points t1 and t2, with corresponding column concentrations for NO2 (denoted as $NO_2^{t1}$ and $NO_2^{t2}$, NOx (denoted as $NO_x^{t1}$ and $NO_x^{t2}$ and their ratios (denoted as $\alpha^{t1}$ and $\alpha^{t2}$), we can calculate the rate of change of NO2 and NOx as $\frac{\partial NO_2}{\partial t}\Big|_{t=t2} = \frac{NO_2^{t2}-NO_2^{t1}}{t2-t1}$ and $\frac{\partial NO_x}{\partial t}\Big|_{t=t2} = \frac{NO_x^{t2}-NO_x^{t1}}{t2-t1}$, respectively. As far as I am concerned, we can only transform the former into something like $\frac{\partial NO_2}{\partial t}\Big|_{t=t2} = \frac{\alpha^{t2} \times NO_x^{t2} - \alpha^{t1} \times NO_x^{t1}}{t2-t1}$, but this doesn't seem equal to $\alpha^{t2} \times \frac{NO_x^{t2}-NO_x^{t1}}{t2-t1} = \alpha^{t2} \times \frac{\partial NO_x}{\partial t}$ unless we assume $\alpha^{t1} = \alpha^{t2}$. Is this assumption being made? Alternatively, can the GEOS-Chem model directly provide $\frac{\partial NO_2}{\partial t}$ and its individual components? I apologize if I've misunderstood any aspects of your methodology.
We account for the temporal variation of alpha. We added the following line: "α(t) is archived every hour for application in Eq. (2)."

9. As you can derive the net change of NO2 from the net change of NOx, would it be more straightforward in Section 4 to directly analyse the individual components of the net change of NO2 rather than combining the net change of NOx with NO2 NOx ratios to facilitate the analyses? Are you doing the latter way because the NO2/NOx ratios may have implications for something like O3 entrainment. Nonetheless, I am assuming that both GEMS and GEOS-Chem can provide O3.
NO and $NO_2$ undergo fast interconversion as we remarked in the main text. We added the following line to further clarify: "It allows us to consider $NO_x$ emission as a source of $NO_2$ even though $NO_x$ is emitted mainly as NO."

10. In lines 249 and 250, why was the presence of the negative transport term linked to the upwind emissions being much lower?
The text is revised as follows: "Even though the emission term remains larger than the chemical loss term, there is also a negative transport term from ventilation."

11. In lines 250 to 255, it appears that you're discussing the variations of NO2 alongside the variations of NOx. Do you have evidence for the maximum concentration of OH at noon, or is this a generally accepted knowledge that's prescribed in the GEOS-Chem model?
OH having the maximum concentration at noon is an accepted knowledge in the atmospheric chemistry field. We added a reference to the referred line as follows: "… reflecting the noon maximum of OH concentrations (Logan et al., 1981) …"

12. In line 258, it appears that the discrepancy between the two Pandora sites is more clearly illustrated in Figure 3 rather than in Figure 2.

Thank you for catching this typo. We replaced Figure 2 with Figure 3 in the text.

13. In lines 260 and 261, could you clarify what range is defined by the Pandora data? Additionally, could you explain how the diurnal variations observed by GEMS and simulated by GEOS-Chem agree within this defined range?
We revised lines 260 – 261 to be "The diurnal variations of GEMS and GEOS-Chem agree to within the ranges defined by data from two Pandora stations" for better clarity.

14. In lines 270 and 271, are you suggesting that if the transport term can be quantified by simple methods, satellite observations could directly indicate the role of emission and chemistry without the need for the GEOS-Chem model? However, the follow-on analysis in Figure 6 still have the transport term under different conditions. Similarly, in lines 293 and 294, are you suggesting that on a regional scale, the tranport term can be marginalized (minimized), leading us to interpret that satellite observations primarily reflect the contributions from emissions and chemistry? How would this statement apply to other regions and periods?
We addressed the above concerns in our response to general comments by Referee #1. We decided to remove these lines.

15. In lines 281 and 282, what is the source of the numerical relationship between wind speed and the ventilation time scale?
The ventilation time scale is defined as $L_x/U$ where $L_x$ is the length in the x-direction and U is the wind speed blowing in the x-direction. We decided to remove the lines 281 and 282 from the main text.

16. In Section 5, I'm curious whether the results for the SMA metropolitan area were obtained simply by averaging the results from the grids that belong to the SMA metropolitan area?
We added the following line in the Figure 7 caption: "Quantities are averages over all $0.25°×0.3125°$ grid cells in the SMA."

17. For Figures 4 — 7, the points on the lines in the first row correspond to exact hours (e.g., 8:00, 9:00, etc.), while the points on the lines in the second and third rows correspond to the half-hour marks (8:30, 9:30, etc.). Is this an intended behavior?
We added the following line to clarify: "Each data point in the second and third rows (centered on the half hour) explains the change between the two successive hours shown in the first row."

**Response to comments by Referee # 1 (Joshua Laughner)**

In "Interpreting GEMS geostationary satellite observations of the diurnal variation of nitrogen dioxide (NO2) over East Asia", the authors compare NO2 column amounts retrieved from the GEMS satellite and surface Pandora measurements with column amounts simulated by GEOS-Chem. They find that the three datasets broadly agree, although better in winter months than summer. They then use tendency diagnostics output by GEOS-Chem to attribute diurnal variations of the NO2 column to emissions, chemistry, and transport. From this, they conclude that diurnal variation of NO2, as observed by GEMS, can be used to estimate NOx emissions in winter months and chemical loss in the summer months, at the scale of an urban area, if the

contribution of transport is accounted for using either a mass balance approach or a chemical transport model.

To me, this paper has the sense of one setting the foundation for future studies. That is fine, and even good to have a relatively short paper focused on demonstrating the necessary foundational concepts, which future papers can point directly to, rather than citing a subsection of larger paper using diurnal information to probe chemical loss or emissions. However, I have one primary concern: it is not clear to me how general these results are. If that concern is addressed, I recommend publication in ACP.

We thank Referee #1 for reviewing our paper and providing helpful comments. Our response to general and detailed comments can be found below:

The crux of my concern is that, as I understood the methodology, this conclusion rests on the chemical loss of NO2 being minimized in winter months, such that the diurnal variation is driven almost entirely by transport and emissions. This does seem likely, at least in mid-latitude cities. However, this may not be true in tropical or sub-tropical areas where winter photochemistry does not decrease as significantly relative to summer. Likewise, this seems to rely on emissions being similar in the winter and summer, such that the diurnal variation in the summer can be decomposed into contributions from the unknown chemical loss rate and the emissions rate inferred from winter data. Figure 7 shows that the changes due to chemical loss and emissions are similar magnitudes, therefore if the emissions were not the same as in the winter, that would introduce considerable uncertainty to chemical lifetimes derived from the diurnal variation in NO2 columns.

I see two routes towards addressing this:

1. The simplest approach would be to explicitly limit the conclusions to Seoul and Beijing. This would still allow this paper to serve as the foundation for future studies of NOx lifetime and emissions for most cities in the GEMS and TEMPO fields of regard, provided the similarity of their winter and summer emissions can be shown.

2. The other route I see would be to expand this manuscript to establish a framework for how to evaluate whether a city would be suitable for this kind of analysis. This would mostly consist of being more explicit about the criteria used to determine whether the necessary conditions are met, e.g.:

- How do you tell that the CTM used in the study is adequately representing transport - is there a confidence level within some observations it must meet?

- How small must the chemical lifetime term in winter be (absolutely or relative to the other terms) such that the winter analysis will produce a good estimate of NOx emissions?

- How similar must winter and summer emissions be to not result in a large uncertainty on the summer chemical lifetimes? How do you test that the emissions are actually this similar, without relying on an emission inventory that may not correctly represent the seasonal variation in NOx emissions?

For this route, it would probably be necessary to find examples of cities that violate these conditions and show that the emissions and lifetimes that would be derived from the diurnal information are incorrect. If this is outside the desired scope, the the first route might make more sense.

Referee #1 makes a good point that simple separation of emissions and chemical loss on the basis of differences in observed diurnal cycles in winter versus summer is fraught with possible errors. This was not intended to be a focus of our paper and we have deleted the corresponding sentence (lines 270-271) and paragraph in the conclusions.

**Specific comments**

l. 123: What about the accuracy of the free troposphere NO:NO2 ratio? The Yang et al. (2023) citation given does seems to examine that and conclude that the model performs reasonably well compared to KORUS-AQ aircraft data. Given that models in the past have exhibited a pernicious bias in the free tropospheric NO:NO2 ratio (e.g. Travis et al., 2016), it would be good to state that explicitly here, if that was indeed the conclusion of Yang et al. (2023).

The line now reads as "Yang et al. (2023) found that the GEOS-Chem simulation during KORUS-AQ successfully reproduced important features of $NO_x$ chemistry, notably the $NO/NO_2$ ratio driven by photochemical cycling involving ozone and $HO_2$."

l. 138: Why did you specifically remove data with quality flag = 12? Is that the standard recommendation in the Pandonia documentation, or did you have a specific reason to do so?

We revised the line as follows: "We exclude low-quality data (quality flag = 12) as recommended by PGN (PGN, 2021)."

l. 141: Is there a specific reason to use GEOS FP over a retrospective product like MERRA-2? It seems like a retrospective met product would remove some uncertainty in met variables for a study using past data.

We used GEOS-FP as it provides higher horizontal resolution (0.25° × 0.3125°) as opposed to MERRA-2 (0.5° × 0.625°). We added this line: "GEOS-FP provides the finest spatial resolution available to drive to GEOS-Chem."

l. 195 and Fig. 4: it might be nice to include a scatter of the GEOS-Chem minus Pandora and GEMS minus Pandora NO2 columns versus hour of day in order to show that GEMS and GEOS-Chem are getting more than just the average diurnal pattern (i.e. is there more scatter in the early morning and late afternoon than near noon, or is the uncertainty pretty consistent). From Fig. 3, it's hard to know what hours of the day the scatter is.

We would rather not add another Figure and it's not clear that much information is to be gained from different levels of scatter at different times of day, considering that useful analysis of diurnal variations focuses on means.

l. 221: I was surprised to see the NO + HO2 pathway but not the NO + RO2 pathway; my initial assumption is that if there is enough HO2 to react with NO, then there would also be RO2 as part of the process of producing HO2. Is it just that, in this environment, most of the RO2 either self-terminates or forms alkyl nitrates instead of generating NO2?

We added NO + $RO_2$ and NO + XO reactions as the new R3 and R4.

l. 250: I'm a little confused by the statement "Without the transport loss term, the NO2 column in summer would still increase over the course of the day." If all you're trying to say is that the net change in NO2 would be positive for all hours of the day without the transport term, I see that. But the net tendency is closer to 0 at the afternoon peak than the morning peak, and the trend into the night looks like it would go up with or without transport. I'd like if this statement could account for those nuances better, or perhaps it would make more sense if it was clearer what conclusion this statement is aimed at supporting.
We removed the line 250.

l. 277: How did you choose 6 m/s as the dividing wind speed?
We added this line to clarify: "A wind speed of 6 m s$^{-1}$ ventilates the $25 \times 25$ km$^2$ urban core on a time scale of one hour."

Fig. 4: Why do panels c and e have one fewer circle than panel a, and likewise for d and f versus b?
We added the following line to clarify: "Each data point in the second and third rows (centered on the half hour) explains the change between the two successive hours shown in the first row."

**References**

Travis et al. "Why do models overestimate surface ozone in the Southeast United States?" 2016, https://doi.org/10.5194/acp-16-13561-2016%

**Response to comments by Referee # 2**

Review of "Interpreting GEMS geostationary satellite observations of the diurnal variation of nitrogen dioxide (NO2) over East Asia" submitted to ACP by Laura Hyesung Yang and colleagues in December 2023.

This paper reports on a detailed analysis of the diurnal cycle in NO2 as observed with the geostationary sounder GEMS over Seoul and Beijing by comparing the satellite-observed winter and summer diurnal variations with those from both a GEOS-CHEM simulation and Pandora ground-based column measurements. From this analysis, the relative importance of emissions, chemical loss, and transport is derived (qualitatively).

In my opinion, the paper is clearly written and based on sound methodology. As one of the first analyses of the diurnal variations in GEMS observations, it definitely deserves publication. Besides the comments and questions already raised in CC1 (by Fei Yao) and RC1 (by Josh Laughner), I have a few additional (minor) questions/comments:
We thank Referee #2 for reviewing our paper and providing helpful comments. Our response to the specific comments can be found below:

Sect. 2.1: GEMS data: Any quality assessment for the official SCDs you can refer to? Some teams decide to not only use their own AMFs, but also redo the DOAS fit (e.g., Lange, K. et al.: Validation of GEMS tropospheric NO2 columns and their diurnal variation with ground-based

DOAS measurements, EGUsphere, https://doi.org/10.5194/egusphere-2024-617, 2024.)
We added the following sentence in Section 2.1: "An alternative DOAS retrieval and AMF by Lange et al. (2024) improved the GEMS L2 $NO_2$ version 2.0 vertical column density product, which was biased due to using incorrect vertical profiles for AMF computation (Oak et al. 2024). Here, we use our own AMF."

Sect. 2.3: You did not consider running the model at a higher spatial resolution, or is that not trivial? I assume you have the emission and meteo data at higher resolution?
We added this line: "GEOS-FP provides the finest spatial resolution available to drive GEOS-Chem."

Sect. 2.3: The need to scale the emission inventory data to better reflect the current situation is clear, but is a country-averaged scaling factor detailed enough? There are probably strong differences between rural and urban NO2 reductions over the past 5-10 years.
We added the following sentences in Section 2.3: "We apply for this purpose the surface $NO_2$ concentration trends for China from the Ministry of Ecology and Environment (MEE) network (MEE, 2023) and for South Korea from the AirKorea network (KEC, 2023), focused mostly on urban sites." and "We assume these scaling factors to be applicable to Beijing and Seoul."

Sect. 2.3: It would have been nice to have/repeat here a figure with the diurnal variation in the AMF. How does its amplitude compare to that in the final VCDs?
Since Yang et al. (2023) already showed a figure with the diurnal variation of the AMF, we referred to the work. We added the following line to Section 2.3: "Yang et al. (2023) showed that GEOS-Chem was successful in reproducing the $NO_2$ vertical profile observed below 5 km altitude and inferred from NO observations above. … The model correctly simulated the observed diurnal variation of the PBL $NO_2$ vertical profile over Seoul as driven by mixed layer growth. This resulted in a diurnal amplitude of 14% for the AMF, peaking in the afternoon when mixing depth is maximum."

Conclusions:

-line 310, "We updated NOx emissions...". In fact, besides the country-wide scaling factor applied to the emission inventory data, you don't write much about the feedback your analysis gives on these inventories. Was the country-wide scaling sufficient?
The country-wide scaling was sufficient for our study (Figure 3). The future works that use the inversion framework would be able to directly provide feedback on the current bottom-up inventories as we reflected on the last line of the Conclusion: "This work further lays the groundwork for use of GEOS-Chem in inversions of the geostationary satellite data to infer $NO_x$ emissions."

-line 314, "Diurnal variation of ..." This only holds for urban, polluted environments. In background conditions, the diurnal cycle of the total column is determined by the stratosphere.
We revised line 314 to be "Diurnal variation of the $NO_2$ atmospheric column in the two cities is mainly determined by the planetary boundary layer (PBL) up to 3 km altitude."